# Investigation on Control Burned of Bagasse Ash on the Properties of Bagasse Ash-Blended Mortars

**DOI:** 10.3390/ma14174991

**Published:** 2021-09-01

**Authors:** Redeat Seyoum, Belay Brehane Tesfamariam, Dinsefa Mensur Andoshe, Ali Algahtani, Gulam Mohammed Sayeed Ahmed, Vineet Tirth

**Affiliations:** 1Department of Materials Science and Engineering, Adama Science and Technology University, Adama 1888, Ethiopia; redeatseyoumseyoum@gmail.com (R.S.); dinsefa.mensur@astu.edu.et (D.M.A.); 2Department of Mechanical Engineering, College of Engineering, King Khalid University, Abha 61413, Asir, Saudi Arabia; alialgahtani@kku.edu.sa (A.A.); vtirth@kku.edu.sa (V.T.); 3Research Center for Advanced Materials Science (RCAMS), King Khalid University, Abha 61413, Asir, Saudi Arabia; 4Department of Mechanical Design and Manufacturing Engineering, Adama Science and Technology University, Adama 1888, Ethiopia; drgmsa786@gmail.com; 5Center of Excellence (COE) for Advanced Manufacturing Engineering, Department of Mechanical Design and Manufacturing Engineering, Adama Science and Technology University, Adama 1888, Ethiopia

**Keywords:** sugarcane bagasse ash, pozzolanic, amorphous silica, OPC and PPC, mortar

## Abstract

In recent years, partial replacement of cement with bagasse ash has been given attention for construction application due to its pozzolanic characteristics. Sugarcane bagasse ash and fine bagasse particles are abundant byproducts of the sugar industries and are disposed of in landfills. Our study presents the effect of burning bagasse at different temperatures (300 °C and 600 °C) on the compressive strength and physical properties of bagasse ash-blended mortars. Experimental results have revealed that bagasse produced more amorphous silica with very low carbon contents when it was burned at 600 °C/2 h. The compressive strength of mortar was improved when 5% bagasse ash replaced ordinary portland cement (OPC) at early curing ages. The addition of 10% bagasse ash cement also increased the compressive strength of mortars at 14 and 28 days of curing. However, none of the bagasse ash-blended portland pozzolana cement (PPC) mortars have shown improvement on compressive strength with the addition of bagasse ash. Characterization of bagasse ash was done using XRD, DTA-TGA, SEM, and atomic absorption spectrometry. Moreover, durability of mortars was checked by measuring water absorption and apparent porosity for bagasse ash-blended mortars.

## 1. Introduction

Partial replacement of cement with waste materials has been given great attention in recent years due to high CO_2_ emissions from cement industries, the high cost of cement, and the need to improve cement properties [1,2]. Typically, ashes of rice husk [3,4], bagasse [5,6,7,8], coffee husk [9], cob corn [10], fly ash [11], and silica fume [12] were studied as their pozzolanic nature has highly reactive amorphous siliceous and aluminous materials. Five (5)% rice husk ash with an average particle size of 95 µm blended OPC have enhanced durability of concrete as well as compressive strength from 36.8 MPa to 38.7 MPa at 28 days of curing [13]. Ash-blended cement also has the potential to reduce the energy consumption of cement manufacturing. Silica fume has also improved high-performance concrete due to its ultra-fine particles, which leads to reducing the porosity of concrete and the formation of calcium silicate hydrate (C–S–H) gel [14,15,16]. Sabir, B.B also investigated high-curing-temperature (50 °C) results in higher strengths of silica-fume-embedded concrete when compared to a lower curing temperature (20 °C) at early ages [14]. Compressive strength, as well as flexural strength of OPC, was enhanced with the addition of silica fume up to 12% [16].

OPC with partial bagasse ash improved the compressive strength compared to ordinary concretes at 28 days curing age [17]. Certain bagasse ash-blended concretes can also enhance the durability of concrete [18] and decrease the heat of hydration [8]. However, industrial bagasse ash has a high carbon content and unburned organic matter [19] which negatively affects concrete properties and also lowers the workability of concrete. Trifunovic, P.D. et al. explained the negative effect of carbon on the compressive strength of bottom ash-blended mortar samples [20]. Thus, we expected an optimum amount of control-burned bagasse ash replacement to have no contrary influence on the properties of Ordinary Portland Cement (OPC).

Therefore, we focused on investigating the BA amount of a control-burned bagasse ash at a high temperature (600 °C/2 h) on the properties of bagasse ash-blended mortar. Mineralogical composition and properties of bagasse ash differ due to diversity of sugar-cane plants, firing temperature and time, cooling rate, quality of bagasse and collection techniques, and ash particle sizes [21]. Specific-gravity and specific-area of BA also differ with firing temperature. Thus, we compared compressive strength and physical properties of bagasse ash-blended OPC/PPC mortar samples with conventional mortar samples. We selected bagasse ash as OPC replacement instead of other waste material ashes because bagasse ash has higher pozzolanic reactivity and is available in large amounts, free of cost in many countries. It was predicted that partial replacement of OPC with well-burnt bagasse ash will increase the compressive strength and durability of blended mortars. 

## 2. Experimental Procedure 

### 2.1. Preparation and Characterization of Bagasse Ash

Bagasse ash (BA) was collected from Wonji-Shoa sugar factory (8°27′14.4″ N, 39°13′48″ E) of Oromia, Ethiopia after burned at a temperature around 300 °C (a lower temperature) and the ash leaves as a landfill as shown in Figure 1a. The dark (char) color of ash was an indication of higher carbon content due to incomplete combustion, which was also explained in previous studies [20,22]. Large amounts of bagasse particles similarly leave as landfills in the factory as shown in Figure 1b. We also collected bagasse particles and then pretreated them in distilled water to remove all undesirable materials. After that, we dried bagasse at 100 °C for 24 h and then burned it at a controlled temperature from 300 °C to 750 °C for 1–2 h (Table 1). Bagasse ash’s color was changed to white as burning temperatures increased due to the elimination of carbon. We analyzed bagasse ash burned at 300 °C (low temperature (LT–BA)) and 600 °C (higher temperature (HT–BA)) using Differential Thermal Analysis-Thermogravimetry Analysis (Shimadzu, DTG-60H). Scanning Electron Microscopy (SEM, Shimadzu, COXIEM-30) and X-ray Diffractometer (Shimadzu, XRD 7000, λ CuK α = 1.5418 Å) were used to confirm the presence of phase and morphology of bagasse ashes. Atomic Absorption Spectrometry (AAS-model, spectra AA-20 plus) was used to analyze bagasse ash’s major and minor oxide compositions. Moreover, we measured the compressive strength bagasse ash-blended mortar samples. Lastly, water absorption and apparent porosity tests for HT–BA blended mortars were performed to check the durability of the specimens according to ASTM C20-00 (2015) [23].

We calcinated bagasse at a controlled temperature from 300 °C to 750 °C for 1–2 h after the bagasse was dried at 100 °C for 24 h. We obtained BA having black to white color/appearance as shown in Table 1. As the burning temperature and duration increase, BA color gradually changed from black to white ash. 

### 2.2. Preparation and Analysis of BA-Blended Mortar Samples

Mortar samples with sizes of 70 × 70 × 70 mm were prepared using bagasse ashes, portland cement, sand, and water as shown in Table 2. Water to cement plus BA (0%, 5%, or 10%) ratio was 0.53. However, as bagasse ash content increased (above 10%), the workability of mixing proportion BA blended mortars decreased a lot. Thus, extra water (20 mL) was added, while BA 15%, 20%, and 25% was added to cement for mortar preparation. The samples were then de-molded after 24 h and measured the compressive strength at the curing age of 3, 7, 14, and 28 days. We used bagasse ash after burning at a high temperature (HT–BA), 600 °C/2 h, and bagasse ash after burning at a low temperature, 300 °C/2 h (LT–BA) as revealed in Table 2 below.

## 3. Results and Discussion 

### 3.1. Mineralogical Compositions of Bagasse Ash (BA)

Table 3 displays mineralogical compositions of BA collected from the sugar industry. We used an Atomic Absorption Spectrometry measurement to reveal the major and minor oxide composition of the bagasse ash, as well as the loss on ignition (LOI) percent to check its pozzolan oxide contents (SiO_2_ + Al_2_O_3_ + Fe_2_O_3_), alkaline oxides, and loss on ignition. The amount of pozzolan oxides (SiO_2_ + Al_2_O_3_ + Fe_2_O_3_) in BA is higher (80.02%) than the minimum content of these oxides (70%) required by ASTM C-618 [24]. The sum of pozzolan oxides of OPC is 31.6%, but bagasse ash has 80.02%, as shown in Table 3. Thus, the result testified the pozzolanic nature of BA as per ASTM C-618 specifications and taken as pozzolan materials for cement replacement. Moreover, loss on ignition (LOI) of BA is 4.75% which is less than 10%. Thus, the pozzolanic activity would upsurge with partial BA replacement cement as stated in a previous study [25]. Chemical analysis data has also indicated BA has a three-times higher silica content (65.06%) than OPC (22.82%). However, bagasse ash has a high alkali content (Na_2_O + K_2_O = 8.66%), implying a high potential for the alkali–silica reaction, which might have an adverse effect.

### 3.2. Thermal Analysis of Bagasse Ash

We used DTA–TGA to measures the thermal stability of bagasse at different temperatures, as illustrated in Figure 2. From the DTA measurement, a small endothermic peak was observed at nearly 70 °C, which is attributed to water evaporation (4.35%). Endothermic peaks between 200 to 300 °C and 350 to 500 °C represent the decomposition of organic matters/compositions of bagasse [27]. During the combustion process, a strong exothermic peak was detected at 356 °C and 446 °C. The first strong peak at 356 °C indicated the oxidation of the volatile product [28]. The second strong absorption peak was at approximately 550 °C, representing carbon removal from bagasse as stated in previous studies; i.e., at nearly 600 °C for prolonged heating, carbon can be removed from BA [19,22]. TGA analysis has shown the mass loss between 200 to 300 °C and 350 to 500 °C is more than 80%, which is caused by the decomposition of organic matter and the combustion of unburned carbon. The second mass-loss is mainly associated with the thermal decomposition of hemicellulose with much less cellulose and lignin, while the third region is especially associated with the cellulose with much less hemicellulose and lignin. 

### 3.3. Compressive Strength of Bagasse Ash (BA)-Blended OPC Mortar Sample

The compressive strength of HT–BA blended and LT–BA blended OPC mortar samples are shown in Figure 3a,b, respectively. A higher compressive strength of mortar was obtained at 5% HT–BA replacement OPC, compared to conventional mortar for all curing ages. The 10% HT–BA replacement also gave high strength to OPC mortars at curing ages of 14 and 28 days (Table 4). Voids among the cement and sand in mortars were probably filled with fine BA, which caused an increase in the BA-blended mortar’s strength at an early age. Moreover, the formation of additional Calcium Silicate Hydrated (C–S–H) gel as a result of the reaction of the active silica of BA and the Ca(OH)_2_ from cement hydration is necessary to improve the strength of mortars. However, the compressive strength of OPC mortars decreased with a high amount of BA replacement at an early age. This is most likely related to the reduction of the 3CaO·SiO_2_ (C_3_S) alite phase amount in a mortar with BA replacement, as the alite phase is responsible for giving early-age strength. Moreover, the 3CaO·Al_2_O_3_ (C_3_A) aluminite phase is also expected to decrease with increasing BA content. Thus, the heat of the hydration rate most probably decreased with BA, as explained in a previous study [8]. In our experimental works, the workability of BA-blended mortar was decreased with increasing BA content, which is related to high water absorption and a larger specific surface area of BA. Thus, we added an extra 20 mL of water as 15%, 20%, and 25% BA-blended mortars were prepared. This might cause higher overall porosity of the mortars. Higher compressive strength was obtained at HT–BA blended OPC mortars compared to LT–BA blended OPC mortars at early ages. 

### 3.4. Compressive Strength of Bagasse Ash-Blended PPC Mortar Samples 

Table 5 illustrates the compressive strength of BA-blended PPC mortars decreased with the increase of BA at early ages up to 28 days. None of the LT–BA and HT–BA addition has shown an improvement in the compressive strength of PPC mortars. This is most likely related to the fact that the alite phase amount decreased with an increasing BA in BA-blended PPC mortars. Nevertheless, the compressive strength variation between PPC mortars and 5% HT–BA-blended PPC mortars was less than 1.5 MPa for all curing ages of 3, 7, 14, and 28 days. The compressive strength of HT–BA- (600 °C/2 h) blended PPC mortars is greater than LT–BA- (300 °C/2 h) blended PPC mortars for all BA replacements and all early curing days, as shown in Table 5. This is probably due to the high carbon amount found at LT–BA. Further studies are required on the effect of HT–BA blended PPC mortars on the properties of mortars at later curing ages. 

### 3.5. Water Absorption and Apparent Porosity of HT–BA-Blended OPC Mortar Samples 

Water absorption of mortars increased from 15.08% to 19.82% with the addition of HT–BA from 0% to 25% at 3 days curing, as shown in Figure 4a, due to a higher specific surface area of bagasse ash (4716 cm^2^/gm), compared to the surface area of cement (3394 cm^2^/g) [29]. BA-blended OPC mortars have higher water absorption at a curing age of 3 days compared to the 7 days of curing for OPC mortars. This may be attributed to an increase in pores spaces in the blocks as the BA percentage increases. Apparent porosity of HT–BA-blended mortars also increased from 22.6% to 26.8% with the increment of BA from 0 to 25% at 3 days curing, as shown in Figure 4b. Apparent porosity is expressed as a percentage of the volume of the internal open pores in the specimen to its exterior volume. 

### 3.6. Microstructure Characterization of Bagasse Ash and Bagasse Ash-Blended Mortar

We prepared HT–BA (bagasse burned at 600 °C) and LT–BA (bagasse burned at 300 °C) for XRD analysis. XRD results for HT–BA illustrate a wider hump between 15° and 30° at 2θ which is an indication of the occurrence of the amorphous silica and also Quartz (Q) and Cristobalite (C) phases (Figure 5). A similar observation was also explained in previous studies [17,30]. This amorphous character contributed to the pozzolanic activity of the material to be added to portland cement. Katare, V.D. et al., stated that the optimal temperature for producing pozzolanic bagasse ash is 600 °C [22]. At this firing temperature of bagasse, it has mainly amorphous silica, which is more reactive and has a high pozzolanic activity index [17,22]. Modification of silica assists in the development of C–S–H, as stated in the previous study [31].

From XRD measurement, we also confirmed the formation of Calcium Silicate Hydrate (C–S–H) in both mortar samples with BA-blended (Figure 6a) and without BA-blended (Figure 6b) mortar. However, BA-blended mortar revealed extra C–S–H peaks at 2θ diffracted angles and more intense C–S–H peaks compared to mortar without BA. This is due to a high proportion of reactive amorphous silica of BA reacted with Ca(OH)_2_ from cement hydration, which is crucial to form C–S–H that assists to increase the strength of samples. In Figure 6a, at 2θ between 25° and 30°, it is shown that CH peak intensity decreased for BA-blended mortar. Similarly, a previous study also used XRD to detect and analyze the C–S–H [32,33]. C–S–H is a nanoscale material, which is mainly responsible for the compressive strength of cement.

SEM images of HT–BA and LT–BA are shown that particles with different sizes and morphology. HT–BA (600 °C/2 h) has porous microstructures (Figure 7a) compared to LT–BA (Figure 7b). It is probably due to the removal of carbon and other components from bagasse ash at a higher temperature (600 °C). We confirmed the effect using DTA–TGA analysis (i.e., at a temperature above 500 °C), and BA mass decreased due to the escape of carbon and other components from BA, as shown in Figure 2. A similar explanation was also provided in previous papers, and they exhibited a decrement of carbon content in sugarcane bagasse ash with an increment of calcination temperature [34]. 

## 4. Conclusions

Based on our study, the following conclusions can be drawn:

Compressive strength ($\sigma_{c}$) of OPC mortar samples increased with an up-to-10% bagasse ash (HT–BA) addition for early curing ages. The strength of mortars increased from 9.26 MPa without BA to 9.40 MPa with HT–BA 5% after 3 days of curing. At 28 days of curing, OPC mortars have $\sigma_{c}$ = 24.5 MPa without BA and $\sigma_{c}$ = 27.3 MPa with BA 5%. Enhancement of the compressive strength of OPC mortars with 5–10% HT–BA are most likely related to extra C–S–H formation in cement paste as a result of reactive amorphous silica of BA reacting with Ca(OH)_2_. It might also relate to the fact that BA probably filled the voids in the mortar. However, the compressive strength of OPC mortars decreased with the addition of BA above 10% at early an curing age, which is likely related to a reduction of the 3CaO.SiO2 (C3S) phase in cement paste. None of the bagasse ash-blended PPC mortars have shown an enhancement in the compressive strength.

Water absorption in bagasse ash-blended OPC mortars increased from 15.08% (without BA) to 19.82% (with the addition of BA 25%) after a 3-day curing age, which might lead to an increase in the pore spaces in dry mortars as the BA percentage increases. When the samples were cured for 7 days, water absorption of BA-blended OPC mortars decreased compared to the samples cured for 3 days. Apparent porosity of BA blended OPC mortars increased from 22% (BA = 0%) to 26.8% (with BA = 25%) after 3 days of curing. 

The pozzolanic activity of bagasse ashes was improved for controlled burning temperature and duration (600 °C/2 h), which facilitated the formation of more C–S–H in mortars. Thus, we improved compressive strength, water absorption, and apparent porosity of bagasse ash-blended OPC mortars by controlling BA amounts, firing temperatures, and duration of bagasse.

## Figures and Tables

**Figure 1 materials-14-04991-f001:**
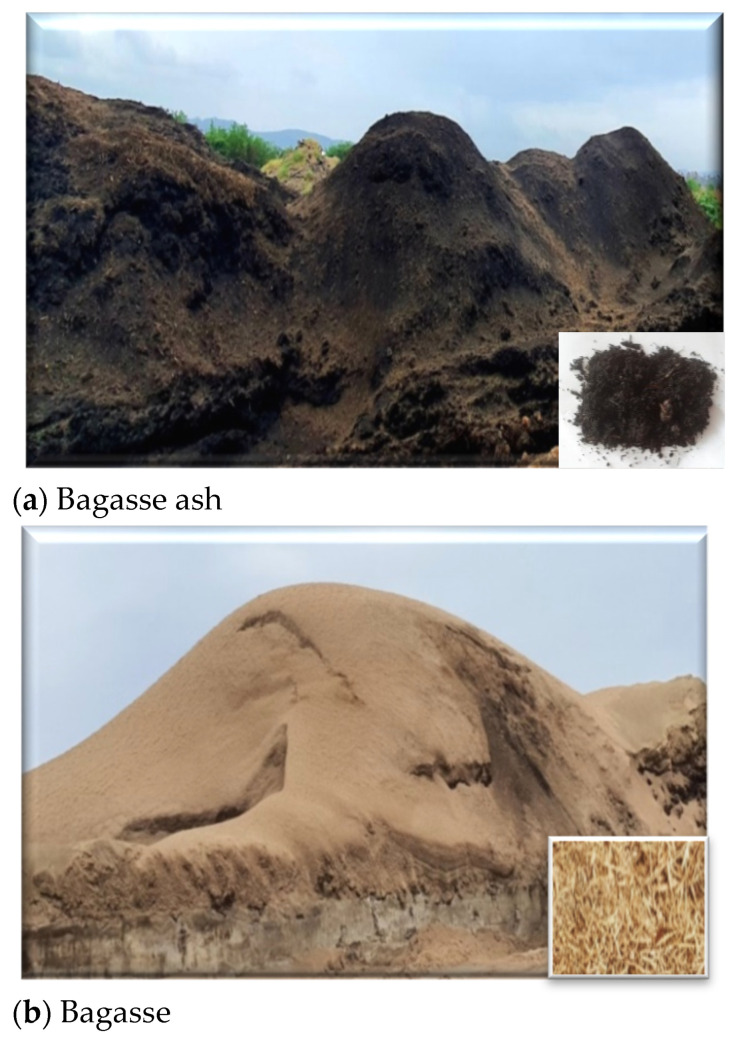
Byproducts as landfills at Wonji Sugar factory, Ethiopia, (**a**) Burned sugarcane bagasse ash at a temperature ≥300 °C for 4–5 min and inset image of bagasse ashes; (**b**) Unburned sugarcane bagasse and inset image of the dry bagasse particles.

**Figure 2 materials-14-04991-f002:**
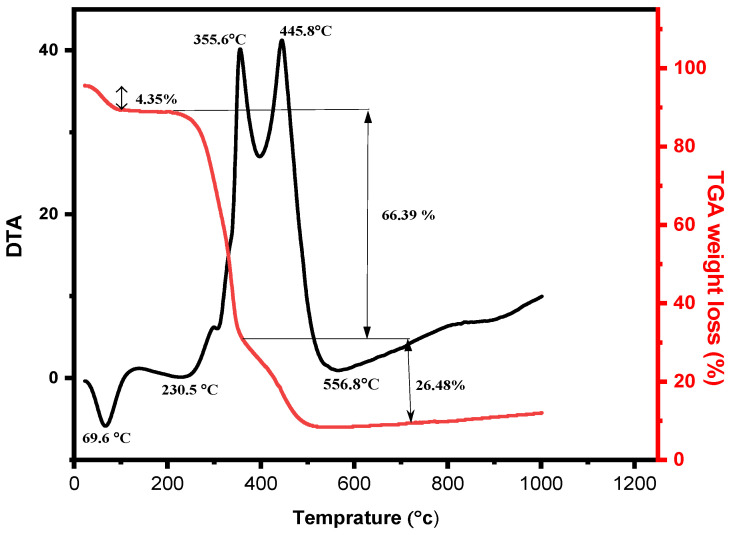
TGA and DTA curves for sugarcane bagasse heating up to 1000 °C.

**Figure 3 materials-14-04991-f003:**
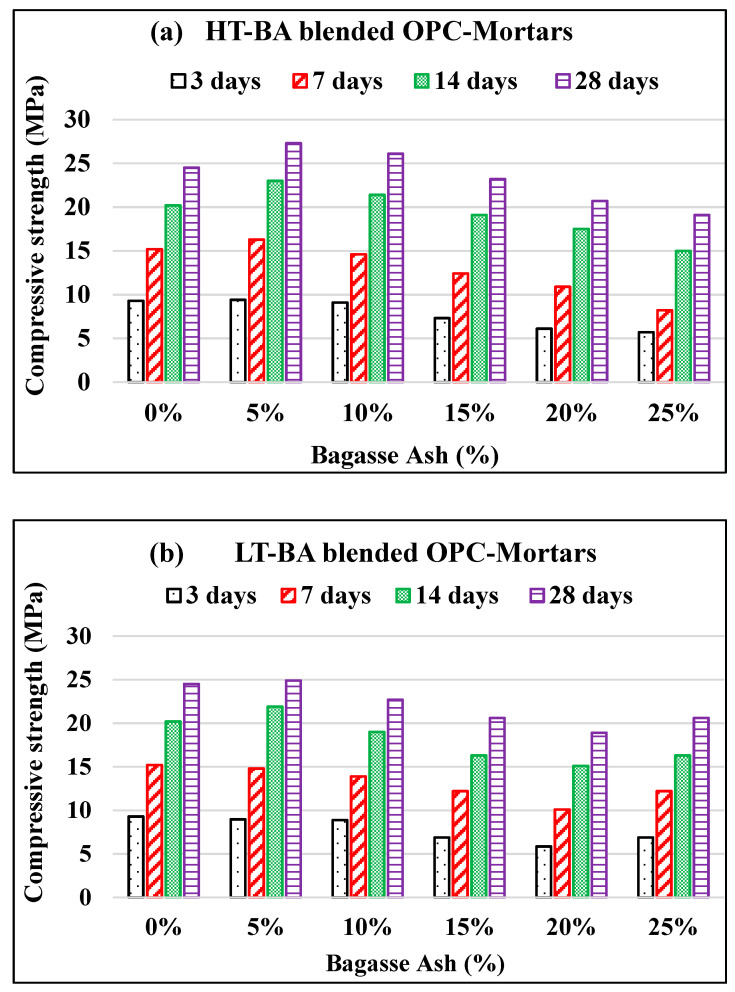
Compressive strength of bagasse ash-blended OPC mortars: (**a**) BA burned at 600 °C/2 h (HT-BA); (**b**) BA burned at 300 °C/2 h (LT-BA).

**Figure 4 materials-14-04991-f004:**
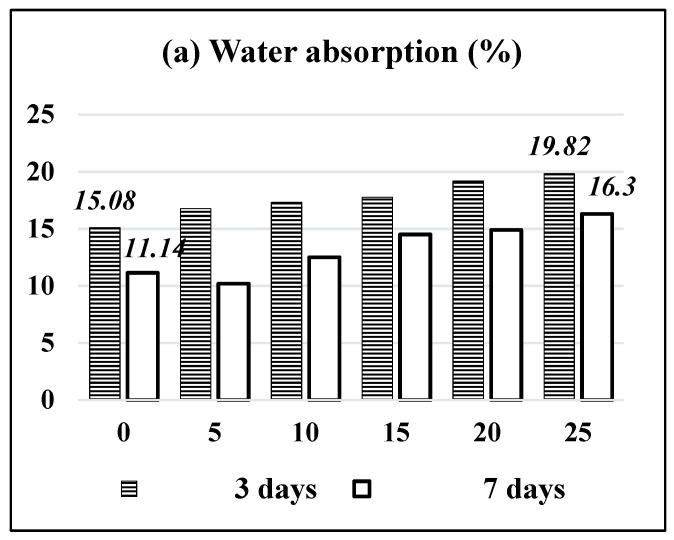
HT–BA blended OPC-mortar samples; (**a**) Water absorption after 3 and 7 days of curing, (**b**) Apparent porosity after 3 and 7 days of curing.

**Figure 5 materials-14-04991-f005:**
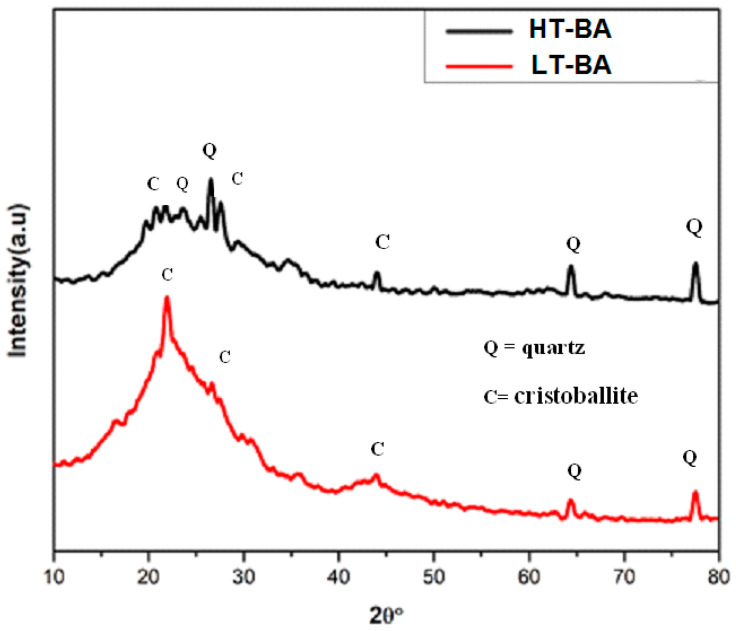
XRD pattern of bagasse burned at 600 °C (top image) and bagasse burned at 300 °C (bottom); Q = Quartz, C = Cristoballite.

**Figure 6 materials-14-04991-f006:**
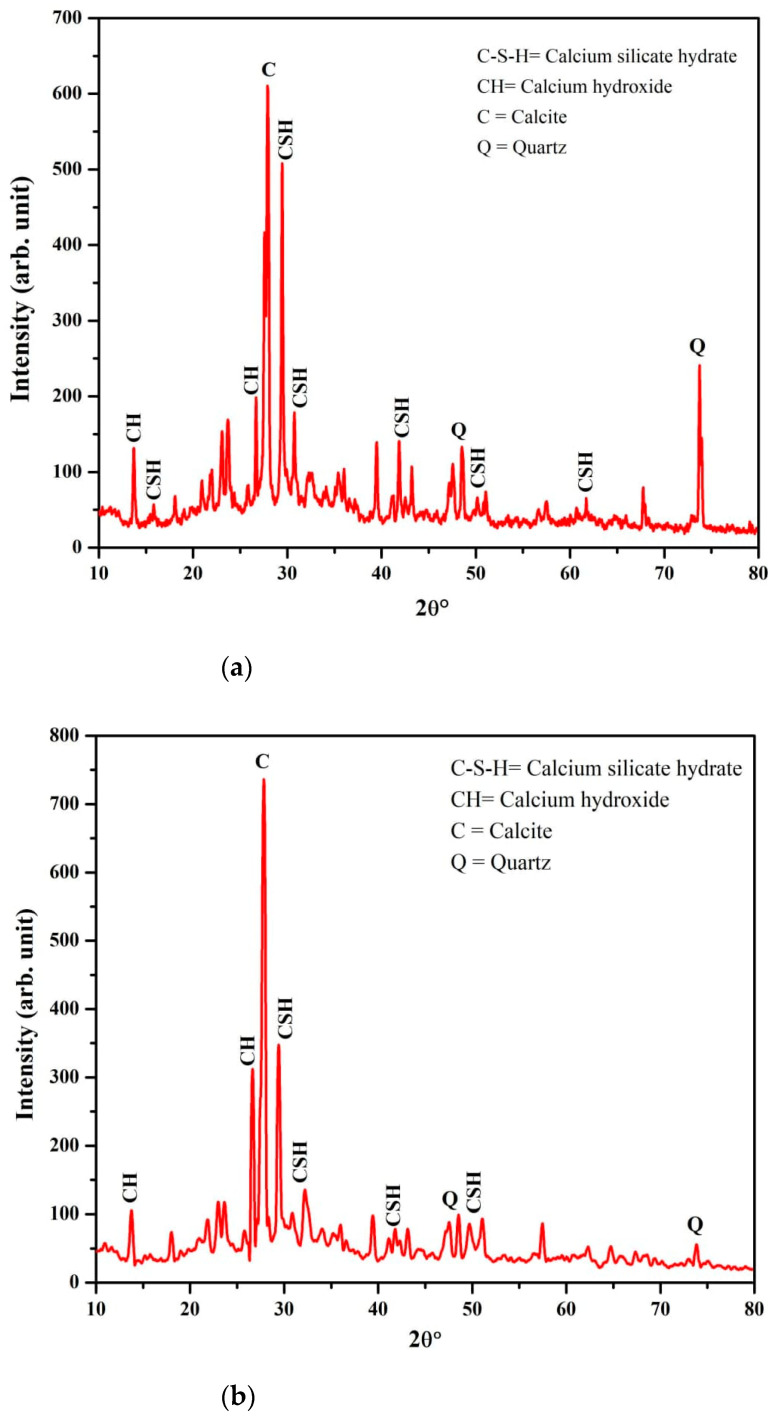
XRD results; (**a**) BA-blended mortar, (**b**) Mortar without BA-blend.

**Figure 7 materials-14-04991-f007:**
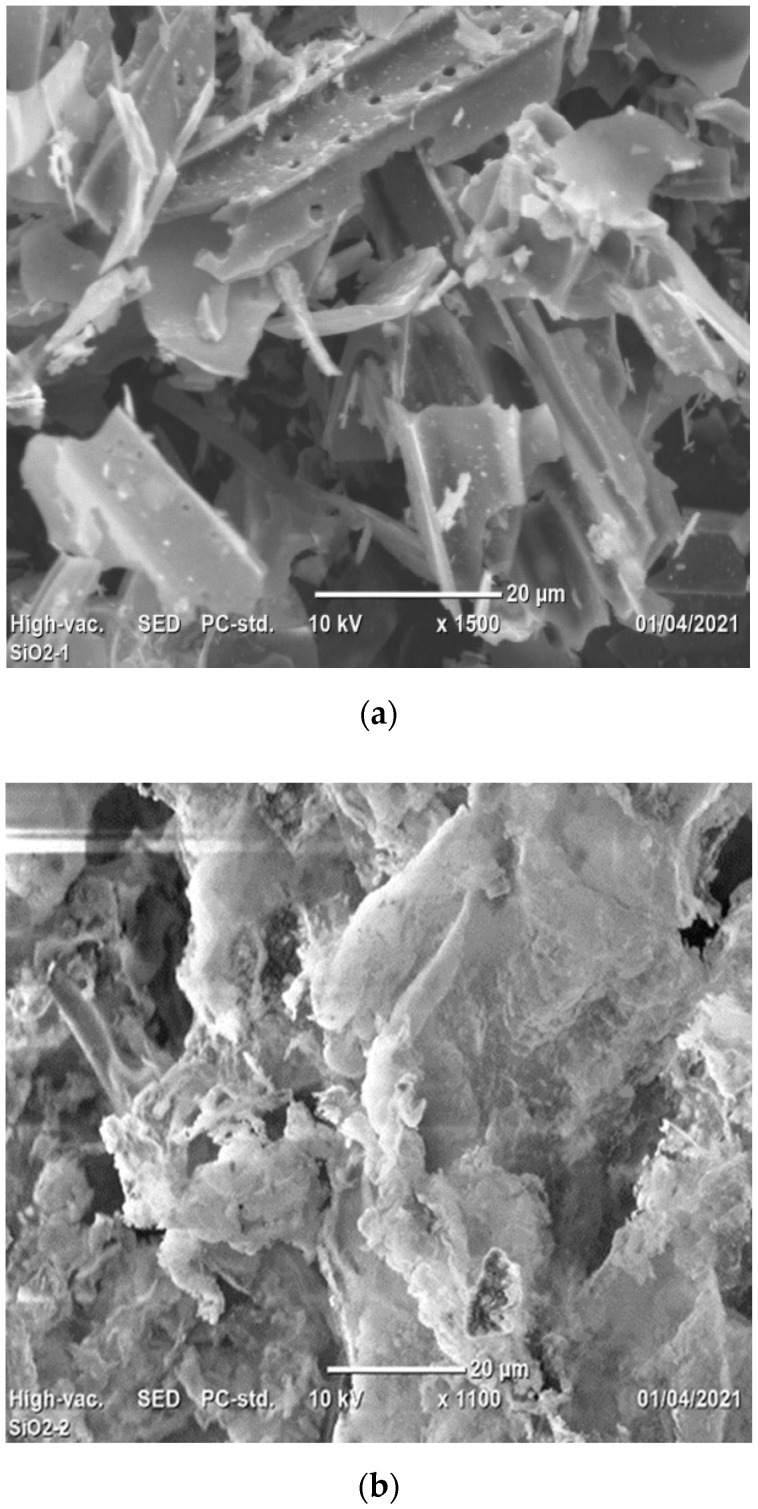
SEM images: (**a**) Controlled-burned (600 °C/2 h) bagasse ash (HT–BA); (**b**) Uncontrolled-burned (300 °C/5 min) bagasse ash (LT–BA).

**Table 1 materials-14-04991-t001:** Bagasse ashes (BA) were prepared by burning 300–750 °C for 1–2 h.

Burning Time	Burning Temperature			
	300 °C	450 °C	600 °C	750 °C
**1 h**	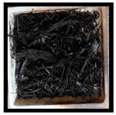	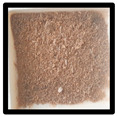	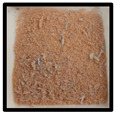	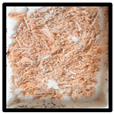
**2 h**	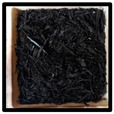	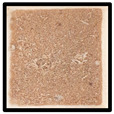	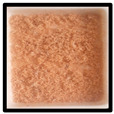	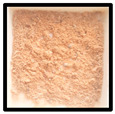

**Table 2 materials-14-04991-t002:** Mixing proportion of BA-blended PC mortars, Cement : Sand = 1 : 3.

Amount BA (%)	PC-Portland Cements (gm)	LT-BA (gm)	HT-B (gm)	Sand (gm)	Water (mL)
BA 0	150.0	0	0	450	80
LT-BA 5	142.5	7.5	0	450	80
LT-BA 10	135.0	15.0	0	450	80
LT-BA 15	127.5	22.5	0	450	100
LT-BA 20	120.0	30.0	0	450	100
LT-BA 25	112.5	37.5	0	450	100
HT-BA 5	142.5	0	7.5	450	80
HT-BA 10	135.0	0	15.0	450	80
HT-BA 15	127.5	0	22.5	450	100
HT-BA 20	120.0	0	30.0	450	100
HT-BA 25	112.5	0	37.5	450	100

**Table 3 materials-14-04991-t003:** Chemical compositions of sugarcane bagasse ash (SCBA) and Dangote OPC cement.

Amount (%)	SiO_2_	Al_2_O_3_	Fe_2_O_3_	CaO	MgO	Na_2_O	K_2_O	MnO	P_2_O_5_	TiO_2_	H_2_O	LOI
Bagasse Ash	65.06	10.88	4.08	1.14	1.30	2.06	6.60	0.10	0.79	0.24	0.66	4.75
OPC [26]	22.82	5.41	3.37	66.32	1.46	-	-	-	-	-	-	-

**Table 4 materials-14-04991-t004:** Compressive strength of bagasse ash (HT–BA and LT–BA) blended OPC mortar samples. BA amount from 0 to 25% of cement replacement.

No.	Bagasse Ash (BA) (%)	Compressive Strength (MPa) of BA Blended OPC–Mortars							
		3 Days		7 Days		14 Days		28 Days	
		LT-BA	HT-BA	LT-BA	HT-BA	LT-BA	HT-BA	LT-BA	HT-BA
1	BA 0	9.26		15.20		20.2		24.5	
2	BA 5	8.97	9.40	14.8	16.29	21.9	23.0	24.9	27.3
3	BA 10	8.88	9.10	13.9	14.61	19.0	21.4	22.7	26.1
4	BA 15	6.89	7.32	12.2	13.50	16.3	19.1	20.6	23.2
5	BA 20	5.85	6.12	10.1	10.90	15.1	17.5	18.9	20.7
6	BA 25	5.30	5.71	7.71	8.21	14.3	15.0	17.30	19.1

**Table 5 materials-14-04991-t005:** Compressive strength of bagasse ash (BA)-blended PPC mortars at early curing ages, 3–28 days; BA amount from 0 to 25% of PPC replacement.

No.	Bagasse Ash (%)	Compressive Strength (MPa) of BA Blended PPC–Mortar Samples							
		3 Days		7 Days		14 Days		28 Days	
		LT-BA	HT-BA	LT-BA	HT-BA	LT-BA	HT-BA	LT-BA	HT-BA
1	BA 0	2.71		3.53		7.05		16	
2	BA 5	1.73	1.83	2.61	3.08	5.10	6.08	12.5	14.7
3	BA 10	1.81	2.34	2.78	3.02	2.42	4.31	6.69	9.36
4	BA 15	1.77	2.04	2.46	2.77	2.12	3.28	4.67	6.91
5	BA 20	1.73	1.79	1.83	1.98	1.48	2.67	4.34	5.95
6	BA 25	1.38	1.42	1.48	1.77	1.44	2.30	2.42	3.97

## Data Availability

The data presented in this study are available on request from the corresponding author.

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
