# Peer review of "Investigation on Control Burned of Bagasse Ash on the Properties of Bagasse Ash-Blended Mortars"

_materials, 2021, doi:10.3390/ma14174991_

Round 1

Reviewer 1 Report

Dear authors,

Some of the objectives in this study were to investigate the optimal temperature and duration of burning material and optimal cement replacement by BA and that was achieved.

Very interesting topic and work, and there is a lot more that can be done.

Here are some questions and suggestions for you.

What was the X-ray source and wavelength for the XRD?

What is the median particle size of this material?

What was the magnification for SEM?

You could determine the CH consumption by using TGA instead of using XRD for qualitative purposes, plus CSH is amorphous phase and is not truly detectable by XRD.

Change CHS to CSH in the figure 7.

You mentioned in the text that this material has high alkalinity and hence would cause ASR issues.

True, but not necessarily if used as SCM. You could also check pH value of mortar.

Strength may not be high, but I am certain this material could find some applications as long as it has good durability properties, so tha tis something to look into, such as ASR.

In general good job.

Author Response

  • Reviewer 1

Comments and Suggestions for Authors

Dear authors,

Some of the objectives in this study were to investigate the optimal temperature and duration of burning material and optimal cement replacement by BA and that was achieved.

Very interesting topic and work, and there is a lot more that can be done.

Here are some questions and suggestions for you.

  • What was the X-ray source and wavelength for the XRD?
  • Response: It is Copper Kα radiation (λCuKα = 1.5418 Å), a scan speed of 4.0 deg/min, 40 kV, 30 mA, and a 2θ scanning range from 10–80°.& we included in the manuscript.
  • What is the median particle size of bagasse ash material?
  • Response: We ground and sieved the bagasse ash using 45m size of sieve (median BA particle size) which is used for our experiment considered
  • What was the magnification for SEM?
  • Response: We used SEM magnifications x1100 & above with the resolution power of 20m since we obtained better images at resolution.
  • You could determine the CH consumption by using TGA instead of using XRD for qualitative purposes, plus CSH is amorphous phase and is not truly detectable by XRD.
  • Response 1: Thank you for your comment. We also measured TGA for cement paste (figure a) & BA blended cement pastes (figure b) to check the thermal stability & CH consumption as shown in the figures below (not included in the manuscript). However, we don’t observe a clear indication which shown the change of CH amount for bagasse ash blended cement paste.
    • Response 2: Probably, in our samples semi-crystalline C-S-H was formed as early age and well-crystalline CSH plates after prolong due to reaction of amorphous silica, CH & H2O as previously study stated analyzing the XRD data [Ref: Baltakys. et.al Influence of modification of SiO2 on the formation of calcium silicate hydrate Materials Science-Poland, Vol. 25, No. 3, 2007].
    • Change CHS to CSH in the figure 7.
    • Response: Thank you so much for your comment: We changed it.
    • You mentioned in the text that this material has high alkalinity and hence would cause ASR issues. True, but not necessarily if used as SCM. You could also check pH value of mortar.
    • Response: thank you so much for your comment. Yes, you are right.

    Our hypothesis was based on chemical compositions analysis looking at alkaline oxide amount (Na2O & K2O) in Sugar cane bagasse ash & Dangote OPC cement. SCBA has higher alkaline oxides contents (8.66%) than Dangote OPC cement which might cause ASR, increase expansion issues. LOI of dangote OPC cement is nearly 1.5%.  Related idea also explained previous study [Simegn A, Abebe S, Worku A (2021) Characterization and Optimization of Incinerated Municipal Solid Waste Fly Ash as a Cement Substitute Material in Concrete at Reppie Waste to Energy Plant, Ethiopia, East Africa. Adv Environ Stud 5(1):382-393] which stated MSWI fly ash was found to have high alkali content (Na2O and K2O) comparable to OPC and it can cause a quick setting, reduce the ultimate strength of concrete, increase expansion underwater, shrinkage under drying conditions and this implies that adding more percentage of incinerated municipal solid waste fly ash to concrete may not be advisable due to its alkali concentration]

    • Strength may not be high, but I am certain this material could find some applications as long as it has good durability properties, so that is something to look into, such as ASR.
    • Response: Thank you so much for your comment and we will check for ASR in the future.

Reviewer 2 Report

The manuscript entitled "Investigation on control burned of bagasse ash on the properties of bagasse ash blended mortars" presents an experimental study conducted on obtaining and characterization cement with treated bagasse ash. However, the introduction section must be improved to provide a clear state of the art in the field of the study, and many other issues must be addressed.

The paper should be rejected.

Some comments follow:

Introduction section

This section must be improved. The citations have been introduced in bulk form “[3-12]” and not distributed in the text following the affirmations that must be supported. Also, to avoid this type of citation, you can cite review studies. Please introduce citation at a specific position to assure a clear correspondence between the affirmations from the introduction section and the previous publication.

In the introduction section, a comprehensive and exhaustive review of the state of the art in the field of the study must be provided. Please refer to previous works, and highlight the experiments and results published previously. In the current form, the introduction section only provides basic/general information about the characteristics of bagasse ashes.

Experimental procedure section

"The Dark (char) color of ash is an indication of higher carbon content due to incomplete combustion." – please cite corresponding studies or provide experimental proof to support your affirmation. Not anything black is reach in carbon.

Table 1. In the provided format it cannot be observed that in Figure 1. a) the microcapsules have different sizes, please introduce scale bar. Also, Figure 1. b) must be improved, it doesn’t have scientific values (please introduce figure caption to indicate the differences and the areas of interest for the reader).

Table 3

The author states that " Table 3 displays mineralogical compositions of BA collected from the sugar industry"- however, in this table only the oxide composition has been presented in this table. Please improve.

Table 3 - - two types of iron oxides have been detected in ash (see DOI: 10.3390/ma13143211), therefore, please replace Fe2O3 with FexOy.

Table 3 – the sum of the elements presented isn’t 100, please check your experimental results.

The DTA-TGA discussions are approximatively and don’t include references from similar studies or proof that the results are correct. Please refer to other studies or databases to confirm the peaks association or include other analysis techniques to prove your results. TG-DTA can be combined with FTIR. Also, the thermal behavior can be correlated with the phases detected by the XRD analysis, by associating the stability or decomposition of specific phases in the temperature range observed and highlighted.

The comparison of the STA results is uncertain. The assignment of weight loss needs stronger evidence.

Also, a clear peak is presented at 250 °C that hasn’t been evaluated by the authors.

" hump between 15o & 30o at 2θ which is an indication of the occurrence of amorphous silica & also quartz (Q) and cristobalite (C) phases (Figure 6). Similarly, observation was also explained in previous studies [17, 24]." This is good, the obtained results are compared and correlated with those presented in other studies. However, the XRD spectra show multiple peaks that haven’t been considered by the author. Could the author explain, why only specific peaks have been considered instead of other? For example, the clear peak around 35-36°, 2θ, is?

"From figure 7 (a & b) XRD results, we can confirm the formation of Calcium Silicate Hydrate (C-S-H)" – this cannot be stated based on the results obtained by XRD. C-S-H formation can be observed by FTIR or other techniques. Please check your experimental results or cite corresponding studies that show a similar procedure/technique.

Mineralogical evaluation. The discussion on XRD patterns is approximative, the identification of some phases is questionable and the evaluation of phase change before and after heating is definitely poor.

"HT-BA has porous structures (figure 8(a)) compared to LT-BA (figure 8 (b))." – figure b shows also porous zones, moreover, this can not be correlated with the C loss. Please see this study to understand amorphous evolution to crystalline phases, the correlation with TG-DTA, other studies, and morphology.

Conclusion:

The conclusion section can be improved since those are far too general.

Conclusions present some of the results discussed above in the paper with very limited discussion.

Reference section

Please check carefully the correlation between the cited papers and the position of that reference in the manuscript text body. Some affirmations have no background in published literature.

Minor observations:

Section numbering – please check, Introduction and Experimental procedures are both section 1

“.” After manuscript title – please remove.

Replace this symbol "&" with and.

"ash which is burned at 300 oC" and "and 600 oC (Higher" – replace 0 with ° degree symbol.

I think that the study needs significant improvement

Author Response

Reviewer 2

Comments and Suggestions for Authors

The manuscript entitled "Investigation on control burned of bagasse ash on the properties of bagasse ash blended mortars" presents an experimental study conducted on obtaining and characterization cement with treated bagasse ash. However, the introduction section must be improved to provide a clear state of the art in the field of the study, and many other issues must be addressed.

Some comments follow:

  • Reviewer Comment 1: Introduction section: This section must be improved. The citations have been introduced in bulk form “[3-12]” and not distributed in the text following the affirmations that must be supported. Also, to avoid this type of citation, you can cite review studies. Please introduce citation at a specific position to assure a clear correspondence between the affirmations from the introduction section and the previous publication. In the introduction section, a comprehensive and exhaustive review of the state of the art in the field of the study must be provided. Please refer to previous works, and highlight the experiments and results published previously. In the current form, the introduction section only provides basic/general information about the characteristics of bagasse ashes.
  • Response: Thank you for your comment. We modified introduction section and citations are placed at specific position as follow:
  1. Introduction

Partial replacement of cement with waste materials has given great attention in recent years due to high CO2 emissions from cement industries, the high cost of cement, and the need to improve cement properties [1,2]. Typically, ashes of rice husk [3,4], bagasse [5-8], coffee husk [9], cob corn [10], fly ash [11], silica fume [12] were studied due to their pozzolanic nature having high reactive amorphous siliceous and aluminous materials. Five (5)% rice husk ash with average particle size of 95m blended OPC-concretes have enhanced compressive strength from 36.8MPa to 38.7MPa at 28 days of curing age & durability of concrete [13]. Ash blended cement has also the potential to reduce the energy consumption of cement manufacturing. Silica fume was also improved the high-performance concrete due to its ultra-fine particles leads to reduce the porosity of concrete & the formation of calcium silicate hydrate (C-S-H) gel [14-16]. Sabir, B.B also investigated at high curing temperature (50 °C) results in higher strengths of silica-fume embedded concrete when compared to lower curing temperature (20 °C) at early ages [14]. Compressive strength, as well as flexural strength of OPC-concrete, enhanced with the addition of silica-fume up to 12% [16].

OPC-Concrete with partial bagasse ash improved the compressive strength compared to ordinary concretes at 28 days curing age [17]. Certain bagasse ash blended concretes can also enhance the durability of concrete [18] and decrease heat of hydration [8]. However, industrial bagasse ash has high carbon content & unburned organic matters [19] which negatively affect concrete properties and also lower the work-ability of concrete. Trifunovic, P.D. et al. explained the negative effect of carbon on the compressive strength of bottom ash blended mortar samples [20]. Thus, we expected optimum amount with control burn bagasse ash replacement has not contrary influence on the properties of OPC-concrete.

Therefore, we focused to investigate the BA amount and a control burned bagasse ash at high temperature (600 °C/2hr) on the properties of bagasse-ash blended mortar. Mineralogical-composition and properties of bagasse ash differ with a diversity of sugar-cane plant, firing temperature & time, cooling rate, quality of bagasse & collection techniques, and ash particle sizes [21]. Specific-gravity and specific-area of BA also differ with firing temperature. Thus, we compared compressive strength & physical properties of bagasse ashes blended-OPC/PPC mortar samples with conventional mortar samples. We selected bagasse ash as OPC cement replacement instead of other waste materials ashes since bagasse ash has higher pozzolanic reactivity and is available in large amounts free of cost in many countries. It was predicted that partial replacement OPC cement with well-burnt bagasse ash will be increased the compressive strength & durability of blended mortars.

  • Experimental procedure section: "The Dark (char) color of ash is an indication of higher carbon content due to incomplete combustion." – Please cite corresponding studies or provide experimental proof to support your affirmation. Not anything black is reached in carbon.
  • Response: Thank you for your comment & we included the citation in the manuscript. We cited two previous studies which stated dark black in ash is an indication of higher carbon content, mainly due to incomplete combustion
  1. Katare, V.D.; Madurwar, M.V. Experimental characterization of sugarcane biomass ash—A review. Constr. Build. Mater. 2017, 152, 1–15.
  2. Qing Xu, Tao Ji, San-Ji Gao, Zhengxian Yang , and Nengsen Wu, Characteristics and Applications of Sugar Cane Bagasse Ash Waste in Cementitious Materials, Materials 2019, 12, 39; doi:10.3390/ma12010039
  • Table 1. In the provided format, it cannot be observed that in Figure 1. a) The microcapsules have different sizes, please introduce a scale bar. Also, Figure 1. (b) Must be improved, it doesn’t have scientific values (please introduce figure caption to indicate the differences and the areas of interest for the reader).
  • Response: Thank you for the comments.
  • We modified figure 1 & write a caption to indicate the differences and the byproducts deposition areas in the manuscript. We also put Inset images for each. Figure 1 (a & b) showed just the photo/appearance of bagasse ash & bagasse particles deposited as a mountain at Wonji Sugar factory, Ethiopia respectively.
  • We changed table 1 naming into Figure (2) looking at the naming way of the previous study for such kind of expression
  • Table 2: The author states that “Table 2 displays mineralogical compositions of BA collected from the sugar industry"- however, in this table only the oxide compositions have been presented in this table. Please improve.
  • Response: We agree with your comment that table 2 has been presented oxide compositions & LOI. We used Atomic Absorption Spectrometry measurement to reveal the major oxides, minor oxide compositions, and Loss of Ignition (LOI) percent of bagasse ash to check its pozzolan oxides contents (SiO2 +Al2O3 +Fe2O3), alkaline oxides, and loss on ignition (LOI) per ASTM 618 standard as explained in previous studies.
  • Table 3two types of iron oxides have been detected in ash (see DOI: 10.3390/ma13143211), therefore, please replace Fe2O3 with FexOy.
  • Response: Thank you for the comment. We checked the paper you suggested (DOI: 10.3390/ma13143211) and observed in fly ash composition wrote as FexOy in place of Fe2O3. I think the paper you mention considers FexOy the same as Fe2O3. However, almost all previous studies (we observed) the ashes including fly ash composition analysis results displayed Fe2O3, not FexOy. For instance, DOI:10.1016/j.cscm.2019.e00263; DOI:1016/j.cemconcomp.2013.10.019; DOI: 10.1016/j.fuel.2012.10.057 etc…
  • Table 2 – the sum of the elements presented isn’t 100, please check your experimental results.
  • Response: Thank you for the comment. We checked the AAS measurement result again, and the results of the composition, in Table 2, do not have errors. However, you are right because the sum of the elements of BA presented isn’t 100. Some oxides like SO3 & others are not presented in table 2, which really occurred in BA. We also confirmed looking at previous studies on the composition analysis of BA. This might be a reason not to get exactly 100%.
  • The DTA-TGA discussions are approximately and don’t include references from similar studies or proof that the results are correct. Please refer to other studies or databases to confirm the peaks association or include other analysis techniques to prove your results. TG-DTA can be combined with FTIR.

Also, the thermal behavior can be correlated with the phases detected by the XRD analysis, by associating the stability or decomposition of specific phases in the temperature range observed and highlighted. A clear peak is presented at 250 °C that hasn’t been evaluated by the authors

  • Response: Thank you for the comment
  1. We included the list of references in our manuscript to prove our DTA-TGA results & explanations. Here are some of the previous studies which supported our methods & results.
  2. [26] Vanita R. Maliger, William O. S. Doherty, Ray L. Frost, and Payam Mousavioun, Thermal Decomposition of Bagasse: Effect of Different Sugar Cane Cultivars, Ind. Eng. Chem. Res. 2011, 50, 791–798
  3. [27]. Garcı`a-Pe`rez, M.; Chaala, A.; Yanga, J.; Roy, C. Co-pyrolysis of sugarcane bagasse with petroleum residue. Part I: thermogravimetric analysis. Fuel 2001, 80, 1245–1258.
  4. [22] Qing Xu, Tao Ji, San-Ji Gao, Zhengxian Yang , and Nengsen Wu, Characteristics and Applications of Sugar Cane Bagasse Ash Waste in Cementitious Materials, Materials 2019, 12, 39; doi:10.3390/ma12010039
  5. Actually, we used dry bagasse raw materials (not bagasse ash) to check the stability or decomposition of specific phases using DTA-TGA analysis. However, for phases detection of the occurrence of amorphous silica & also quartz (Q) and cristobalite (C) phases in bagasse ash using XRD analysis.
  • In our manuscript, We explained Endothermic peaks between 200 – 300  indicate the decomposition of compositions of bagasse as stated previous study [27]
  • “XRD Hump between 15o & 30o at 2θ which is an indication of the occurrence of amorphous silica & also quartz (Q) and cristobalite (C) phases (Figure 6). Similarly, observation was also explained in previous studies [17, 31]." This is good, the obtained results are compared and correlated with those presented in other studies. However, the XRD spectra show multiple peaks that haven’t been considered by the author. Could the author explain why only specific peaks have been considered instead of others? For example, the clear peak around 35-36°, 2θ, is?
  • Response: We have considered specific peaks show the occurrence of amorphous silica & also quartz (Q) and cristobalite (C) phase, specifically amorphous silica which is important to confirm our experimental results. Amorphous character contributed to the pozzolanic activity of the material to be added to Portland cement.
  • "From figure 7 (a & b) XRD results, we can confirm the formation of Calcium Silicate Hydrate (C-S-H)" – this cannot be stated based on the results obtained by XRD. C-S-H formation can be observed by FTIR or other techniques. Please check your experimental results or cite corresponding studies that show a similar procedure/technique.
  • Response: Thank you for the comment. We included the following references in our manuscript to verify the similar procedure/technique used to confirm the formation of CSH using XRD Analysis.
  1. Nasir, M.; Al-Kutti, W.; Performance of Date Palm Ash as a Cementitious Materials by Evaluating Strength, Durability, and Characterization. Buildings. 2019, 9 (6), 1-13.
  2. Bergold, S.T.; Goetz-Neunhoeffer, F.; Neubauer, J.; Quantitative analysis of C–S–H in hydrating alite pastes by in-situ XRD. Cem. Concr. Res. 2013, 53, 119–126.

 Our XRD results (Figure 7) shown that the formation of C-S-H in both BA-blended mortar samples & mortar without BA-blended. Both samples have intensity associated with C-S-H. However, BA-blended cement pastes show additional C-S-H at different diffracted angles and more intense C-S-H peaks found. CH peak intensity also decreased for BA blended mortar. This is due to extra C-S-H formation with the addition of BA by reaction of CH & SiO2 of BA. A similar previous study also used XRD to confirm the formation of CSH with the addition of BA.

  • Mineralogical evaluation. The discussion on XRD patterns is approximative, the identification of some phases is questionable and the evaluation of phase change before and after heating is definitely poor.
  • Response: we supported our XRD results for phase identification is supported by previous studies. We checked the phases formed after bagasse ash (prepared by burning at 300 & 600   as shown in figure 6 using XRD. At 600 oC the firing temperature, bagasse ash has mainly amorphous silica which is more reactive, similarly stated in a previous paper [31].

We also confirmed the formation of Calcium Silicate Hydrate (C-S-H) in both mortar samples with BA-blended & without BA-blended using. We didn’t check at the constituent in the bagasse before heating since it is known to dry bagasse majorly consists of cellulose, hemicellulose, and lignin.

  • "HT-BA has porous structures (figure 8(a)) compared to LT-BA (figure 8 (b))." – figure b shows also porous zones, moreover, this can not be correlated with the C loss. Please see this study to understand amorphous evolution to crystalline phases, the correlation with TG-DTA, other studies, and morphology.
  • Response: Thank you for the comment
  1. Appearance/color change of BA with increasing burning temperature is one of an indication of escaping of Carbon from BA as it also stated in previous studies (we included the references in the manuscript as evidence).
  2. We measured BA using DTA-TGA. At temperature above 500 oC, BA mass decreased due to carbon escaping from BA(figure 3) referring to previous studies

Thus, these results supported the SEM images results (figure 8). SEM image 8(a), BA prepared after burning at 600 C (HT-BA). It consists of porous morphology compared to 300 oC (Figure 8b). Based on our experimental results & referring to previous studies, porous micromorphology in SEM image is most likely relate to escaping of carbon & other components at higher burning temperature (600 oC) & longer duration (2hr)

  • Conclusion: …..present some of the results discussed above in the paper with very limited discussion.
  • Response: Thank you for the comment. We modified the conclusion section as follow

Based on our study, the following conclusions can be drawn:

Compressive strength () of OPC-mortar samples increased up to 10% bagasse ash (HT-BA) addition for early curing ages. The strength of mortars increased from 9.26MPa without BA to 9.40MPa with HT-BA 5% after 3 days of curing. At 28 days of curing, OPC-mortars have=24.5MPa without BA and =27.3MPa with BA 5%. Enhancement of compressive strength of OPC-mortars with 5-10% HT-BA are most likely related to extra C-S-H formation in cement paste as a result of reactive amorphous silica of BA reacted with Ca(OH)2. It might also relate to BA probably filled the voids in mortar. However, compressive strength of OPC-mortars decreased with the addition of BA above 10% at early curing age which is likely related to the reduction of 3CaO.SiO2 (C3S) phase in cement paste. None of the bagasse ash blended PPC mortars have shown an enhancement in the compressive strength.

Water absorption bagasse ash blended OPC-mortars increased from 15.08% (without BA) to 19.82% (with the addition of BA 25%) after 3-day curing ages which might be lead to an increase in the pores spaces in dry mortars as the BA percentage increases. When the samples were cured 7 days, water absorption of BA-blended OPC-mortars decreased compare to the samples cured 3 days. Apparent porosity of BA blended OPC-mortars increased from 22% (BA=0%) to 26.8% (with BA=25%) after 3 days of curing.

The pozzolanic activity of bagasse ashes was improved for controlled burning temperature & duration (600 °C/2hr) which facilitated the formation of more C-S-H in mortars. Thus, we improved compressive strength, water absorption, and apparent porosity of bagasse ash blended OPC- mortars by controlling BA amounts, firing temperatures & duration of bagasse. 

  • Reference section
  • Response: Thank you for the comment. We crosschecked each cite and included some references as proof.
  • Minor observations:
  • Response: we modified it per your comments.

Round 2

Reviewer 2 Report

The manuscript was improved in accordance with the suggestions. However, there are still some issues that must be addressed that can improve the clarity of the manuscript.

The responses and explanations regarding the chemical composition and XRD analysis are apposite. However, these comments should be briefly introduced in the manuscript, in order to make the results clear, also, for the reader.

Author Response

Response to Reviewer 2

Point: The manuscript was improved in accordance with the suggestions. However, there are still some issues that must be addressed that can improve the clarity of the manuscript.

The responses and explanations regarding the chemical composition and XRD analysis are apposite. However, these comments should be briefly introduced in the manuscript, in order to make the results clear, also, for the reader.

ʉ۬Response 1: Thank you so much for your comment. We briefly incorporated the explanation regarding the chemical composition and XRD analysis in the manuscript. However, some of the following points are already introduced in the manuscript.

  • Chemical composition analysis revealed the major oxides, minor oxide compositions, and Loss of Ignition (LOI) percent of bagasse ash to check its pozzolan oxides contents (SiO2 +Al2O3 +Fe2O3), alkaline oxides, and loss on ignition (LOI) per ASTM C-618 standard.
  • XRD analysis results for HT-BA illustrate a wider hump between 15o & 30o at 2θ which is mainly an indication of the occurrence of amorphous silica & also quartz (Q) and cristobalite (C) in bagasse ash phases as shown in figure 6. A similar observation was also explained in previous studies [17,30]. At 600 ℃ firing temperature, bagasse ash has mainly amorphous silica which is more reactive, similarly stated in the previous paper [31].
  • The formation of Calcium Silicate Hydrate (C-S-H) is also checked in both mortar samples with BA-blended & without BA-blended using XRD (figure 7)….
